# Collateral sensitivity constrains resistance evolution of the CTX-M-15 β-lactamase

Carola E.H. Rosenkilde[1], Christian Munck[1,3], Andreas Porse[1], Marius Linkevicius[2], Dan I. Andersson [2] & Morten O.A. Sommer [1]

Antibiotic resistance is a major challenge to global public health. Discovery of new antibiotics is slow and to ensure proper treatment of bacterial infections new strategies are needed. One way to curb the development of antibiotic resistance is to design drug combinations where the development of resistance against one drug leads to collateral sensitivity to the other drug. Here we study collateral sensitivity patterns of the globally distributed extended-spectrum β-lactamase CTX-M-15, and find three non-synonymous mutations with increased resistance against mecillinam or piperacillin–tazobactam that simultaneously confer full susceptibility to several cephalosporin drugs. We show in vitro and in mice that a combination of mecillinam and cefotaxime eliminates both wild-type and resistant CTX-M-15. Our results indicate that mecillinam and cefotaxime in combination constrain resistance evolution of CTX-M-15, and illustrate how drug combinations can be rationally designed to limit the resistance evolution of horizontally transferred genes by exploiting collateral sensitivity patterns.

---

[1] Novo Nordisk Foundation Center for Biosustainability, Technical University of Denmark, DK-2800 Lyngby, Denmark. [2] Department of Medical Biochemistry and Microbiology, Uppsala University, Box 582, SE-751 23 Uppsala, Sweden. [3] Present address: Department of Systems Biology, Columbia University, New York, NY, USA. These authors contributed equally: Carola E. H. Rosenkilde, Christian Munck. Correspondence and requests for materials should be addressed to M.O.A.S. (email: msom@bio.dtu.dk)

Antibiotics are essential to modern medicine but the introduction of new antibiotics is inevitably followed by the emergence of antibiotic-resistant bacteria as a result of either chromosomal mutations (adaptive evolution) or horizontal gene transfer (HGT)[1–3]. The emergence of resistance, in combination with the limited development of new drugs, has led to a marked reduction in our ability to treat bacterial infections efficiently[4–6]. Accordingly, there is a growing interest in using existing antibiotics to develop treatment strategies that both eliminate the unwanted bacteria and extend the life span of existing antibiotics[7–9].

One such strategy is antibiotic combination therapy, which can both increase the bacterial target spectrum to include resistant variants and prevent emergence of resistance. Combination therapy has been successfully applied against infections since the 1940s and has improved the outcome of diseases such as tuberculosis and HIV[10–16]. However, it is a challenge to combine drugs that not only have high potency against the pathogen but also constrain the evolution of resistance, owing to our limiting understanding of phenomena such as positive and negative drug interactions (synergy and antagonism), as well as collateral resistance and sensitivity[17–19].

Numerous studies have examined how drug pairs can be rationally designed. Previously, we and others have shown that antibiotic combinations, in which the evolution of resistance to one of the drugs leads to collateral sensitivity against the other drug, effectively constrain the evolution of adaptive resistance in *Escherichia coli*[20], *Staphylococcus aureus*[21,22] and *Pseudomonas aeruginosa*[23]). Such studies motivated the development of mathematical[24] and computational models, such as flux balance analysis[25] and fitness landscape analysis of the evolutionary potential of resistance[26,27], which may guide in the development of rationally designed drug combinations.

A common approach for studying resistance evolution and rational drug design is to use adaptive evolution to identify mutations conferring increased antibiotic resistance. Such studies have led to the identification of drug combination regimens proposed to limit resistance evolution. However, this effect will often depend on the nature of the interaction between the two drugs and the targeted organism[17,18,28,29]. Furthermore, adaptive resistance evolution does not consider the effect of horizontally transferred genes, which constitute a major source of antibiotic resistance for many pathogens[30] and therefore are important to study in relation to the efficacy of drug combinations[1,9]. Some of the studies that have been performed on horizontally transferred genes aimed to quantify and predict the resistance evolution by applying methods such as directed evolution using error-prone PCR[31,32], as well as adaptive evolution methods[33–36]. As an example, treatment strategies that apply a selective pressure to revert extended-spectrum-resistant genotypes of TEM β-lactamases back to the Wild Type (WT) state have been proposed by a data-driven model based on the fitness costs of specific resistance gene variants[37].

Another way to study resistance evolution and to apply rational drug design of horizontally transferred resistance genes is to elucidate collateral sensitivity patterns by using a random mutagenesis approach. The benefit of using a random mutagenesis approach to study evolution of horizontally transferred genes is threefold: (1) It is an effective way of mimicking the evolutionary process assuming that one mutation is present in each clone. (2) A large library can easily be screened on various antibiotics. (3) Traditional adaptive laboratory evolution methods are not always suitable to study evolution of single horizontally transferred genes due to the potential appearance of mutations in the host genome that confer to increased resistance towards the specific antibiotic[38].

We use this method to study resistance evolution in the horizontally transferred β-lactamase gene $bla_{CTX-M-15}$, which represents the most common variant of the large CTX-M family of extended-spectrum β-lactamases (ESBLs). $bla_{CTX-M-15}$ is globally disseminated and represents a substantial challenge in the treatment of multidrug-resistant *E. coli* infections[39,40]. Bacteria harbouring CTX-M-15 are highly resistant to β-lactams, including penicillins and cephalosporins[41]. However, they are commonly susceptible to the β-lactam drugs: mecillinam, meropenem, and piperacillin in combination with the β-lactamase inhibitor tazobactam (piperacillin–tazobactam) (Supplementary Table 1)[42].

Here we examine whether single mutations in the $bla_{CTX-M-15}$ gene will provide increased resistance towards any of these drugs and elucidate how collateral sensitivity can be exploited to propose drug combinations that constrain the resistance evolution of $bla_{CTX-M-15}$. We find three single mutations that increase either mecillinam or piperacillin resistance, while simultaneously decreasing cephalosporin resistance. We verify in vitro and in mice the effect of the drug combination mecillinam and cefotaxime as effective in eliminating both the CTX-M-15WT and the resistant mutant. Based on this, we propose that a drug combination of mecillinam and cefotaxime can restrain resistance evolution in CTX-M-15, and we elucidate how collateral sensitivity patterns of horizontally transferred genes can be exploited for rational design of antibiotic combinations.

## Results

**Discovery of three highly resistant CTX-M-15 mutants.** To examine whether any single mutations of the $bla_{CTX-M-15}$ gene could lead to increased resistance against the β-lactam drugs mecillinam and meropenem or the β-lactam-inhibitor combination piperacillin–tazobactam, we generated a library of $bla_{CTX-M-15}$ mutants using error-prone PCR. The library was transformed into *E. coli* TOP10 and the transformation was selected on plates containing different concentrations of the tested drugs, separately. The cephalosporin ceftazidime, an antibiotic to which CTX-M-15 confers high resistance, was used as a positive control. Deep sequencing of the $bla_{CTX-M-15}$ library was performed for each selection plate (colony-forming unit (CFU) count is in Supplementary Tables 2 and 3). Sequencing of the library plated on Lysogeny Broth (LB) medium without antibiotics was performed to assess the background mutation distribution (Supplementary Tables 2 and 3). To detect mutants that had gained both low- and high-level resistance, screening was performed at concentrations below and above the minimal inhibitory concentration (MIC) of the antibiotics tested (0.5–4× MIC, Table 1 in Methods) (Fig. 1a–c). We estimate an average coverage of each possible mutation in the $bla_{CTX-M-15}$ library to be 280-fold (Methods).

Three highly resistant mutants were found: two on the piperacillin–tazobactam selection plate (CTX-M-15$_{S133G}$ and CTX-M-15$_{G239S}$) and one on the mecillinam plate (CTX-M-15$_{N135D}$). The frequency of these mutants within the total population (referred to as the total population frequency) was highest on the higher-concentration selection plates, indicating that they have a selective advantage over less resistant clones. Specifically, the mutants CTX-M-15$_{S133G}$ and CTX-M-15$_{G239S}$ were identified with a total population frequency of 25% and 16%, respectively, on the 4× MIC piperacillin–tazobactam plates, and the CTX-M-15$_{N135D}$ mutant was identified with a total population frequency of 52% on the 4× MIC mecillinam selection plates. In addition, a synonymous mutation, assumed to be linked to one of the CTX-M-15$_{N135D}$ mutants, was observed at position 91 with a frequency of 9% at 4× MIC of mecillinam (Fig. 1a and b).

None of the clones showed increased resistance against meropenem (Fig. 1c). In addition, we screened the library on

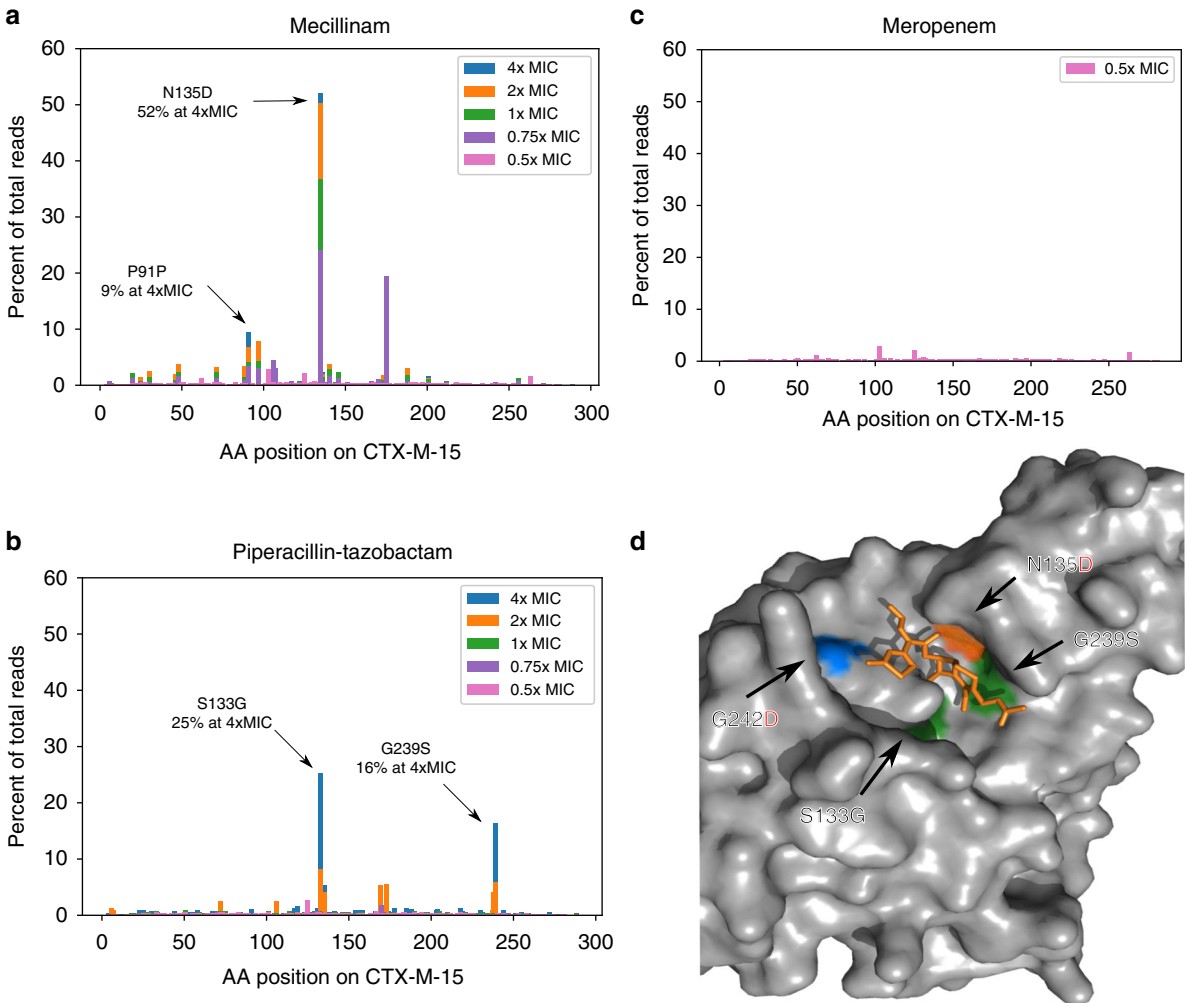

**Fig. 1** Frequency of CTX-M-15 clones selected on antibiotics and position of mutations. A blaCTX-M-15 gene mutant library obtained by error-prone PCR was grown on different antibiotic concentrations (0.5–4 × minimal inhibitory concentration (MIC)), the resulting clones were sequenced, and the percentage of reads for each SNP is shown for: mecillinam (**a**), piperacillin–tazobactam (**b**), and meropenem (**c**). The CTX-M-15 wild-type (CTX-M-15 WT) MIC for mecillinam is 0.25 μg/ml, for piperacillin–tazobactam is 16 μg/ml, and for meropenem is 0.06 μg/ml. **d** A 3D structure of the CTX-M-15-binding site with cefotaxime (orange) (PDB: 5FAP [https://www.rcsb.org/structure/5FAP]). Mutations providing high resistance against mecillinam: N135D (orange), against piperacillin–tazobactam: S133G and G239S (green), and amoxicillin–clavulanic acid (blue). Amino acid (AA) letters with uncharged side chains are written in black and negatively charged side chains in red

the β-lactam-inhibitor combination amoxicillin–clavulanic acid, which CTX-M-15 provides resistance above the clinical breakpoint of 8 μg/ml (defined by the European Committee on Antimicrobial Susceptibility Testing (EUCAST)). The MIC of CTX-M-15$_{WT}$ is 14 μg/ml (Supplementary Table 1). We did this to determine whether it was possible for CTX-M-15 to increase resistance against this drug combination with only one mutation. One mutant, which was selected on amoxicillin–clavulanic acid (32 μg/ml), CTX-M-15$_{G242D}$, showed increased resistance against amoxicillin–clavulanic acid. As CTX-M-15$_{WT}$ already provides resistance against this drug, the mutation has limited impact in vivo. Therefore, this mutant was not included in subsequent experiments.

By visualizing the resistance mutations on a three-dimensional structure of the CTX-M-15 enzyme bound to the cephalosporin cefotaxime (Fig. 1d), it was clear that all amino acid substitutions were clustered in the active site of the enzyme and therefore were likely to change the binding affinity for the three investigated drugs as well as for cephalosporins.

**CTX-M-15 mutants exhibit collateral sensitivity to cephalosporins.** To characterize the changes in antibiotic susceptibility of the mutants, as well as possible epistatic interactions between them, single and double mutants were constructed and expressed in *E. coli* TOP10. Antibiotic susceptibility was determined for a panel of β-lactam antibiotics towards which CTX-M-15$_{WT}$ confers resistance: amoxicillin–clavulanic acid and cefotaxime, as well as some β-lactams to which CTX-M-15$_{WT}$ does not confer resistance: mecillinam alone and in combination with the inhibitors clavulanic acid or tazobactam, meropenem, and piperacillin alone, and in combination with tazobactam. Mecillinam was tested together with the inhibitors to test for drug synergy or antagonism. The MIC fold changes were calculated for each drug and each mutant relative to CTX-M-15$_{WT}$ (Fig. 2a and b).

The single mutant selected on mecillinam, CTX-M-15$_{N135D}$, showed a 50-fold increase in mecillinam MIC compared with CTX-M-15$_{WT}$ (from 0.3 μg/ml to 15 μg/ml), thereby exceeding the clinical breakpoint of 8 μg/ml (as defined by EUCAST) (Supplementary Table 1). When exposed to mecillinam in

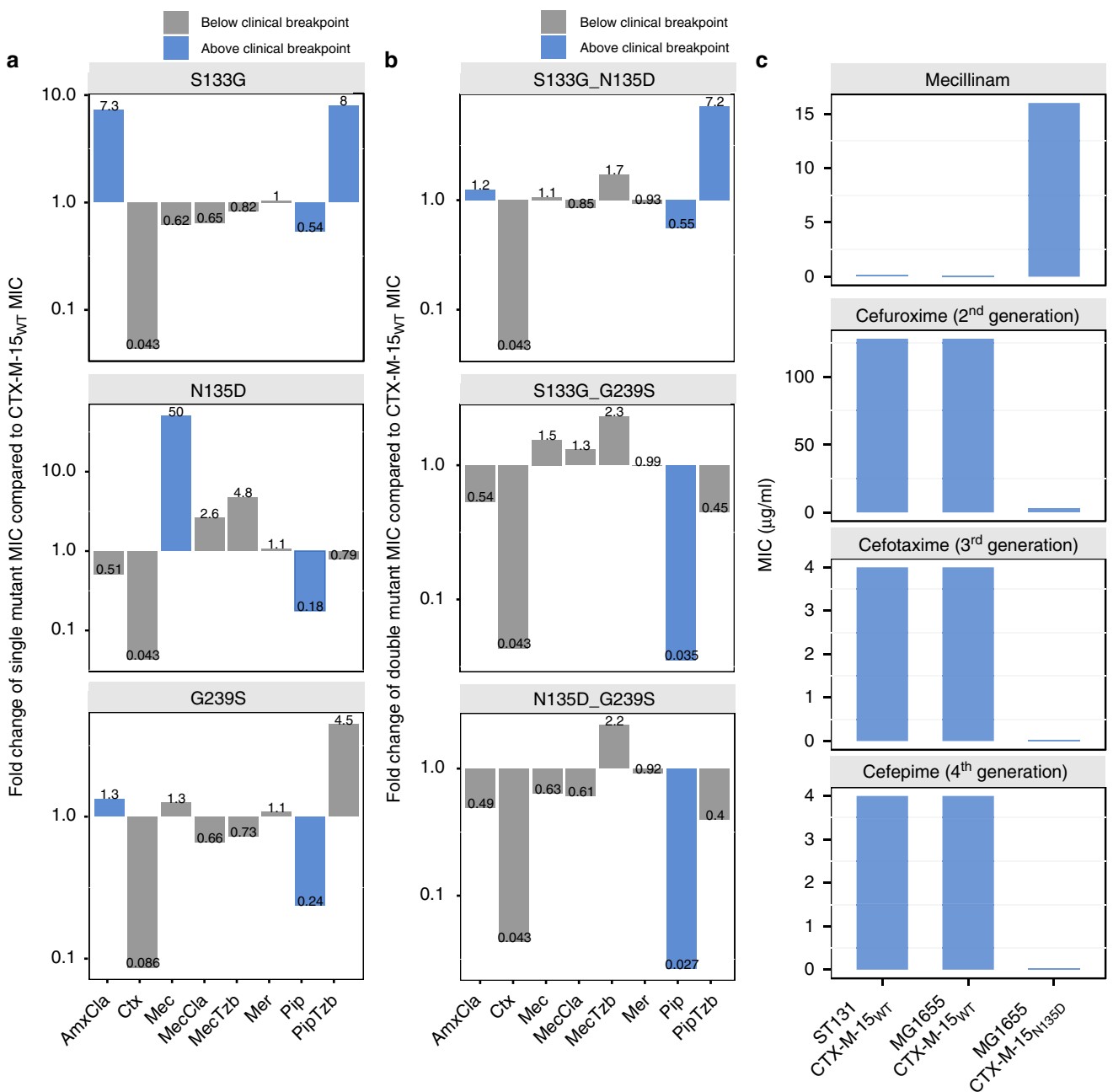

**Fig. 2** Fold change of re-constructed mutants and MIC values for different CTX-M-15 carrying strains. **a** Fold change of the MIC of the indicated antibiotics for *E. coli* TOP10 expressing the indicated single mutants of CTX-M-15. Substantial collateral sensitivity and resistance results from individual mutants selected in specific antibiotics. **b** Fold change of the MIC of the indicated antibiotics for *E. coli* TOP10 expressing double mutants of CTX-M-15. Negative epistasis is observed for all double mutants except CTX-M-15S133G_N135D towards piperacillin–tazobactam. Fold changes for single and double mutants are in relation to the MIC values of *E. coli* TOP10 expressing CTX-M-15WT. **c** The MIC was determined for the mecillinam-resistant mutant CTX-M-15N135D against mecillinam and three different cephalosporins: cefuroxime (second generation), cefotaxime (third generation), and cefepime (fourth generation). Two different strains were used as wild-type (WT) controls of CTX-M-15: *E. coli* MG1655 and a clinical *E. coli* ESBL strain, ST131. *E. coli* MG1655 was used for expression of CTX-M-15N135D. Blue bars indicate MIC values above the clinical breakpoint (as defined by the European Committee on Antimicrobial Susceptibility Testing (EUCAST) (Supplementary Table 1). AmxCla, Amoxicillin–clavulanic acid; Ctx, cefotaxime; Mec, mecillinam; MecCla, mecillinam–clavulanic acid; MecTzb, mecillinam–tazobactam; Mer, meropenem; Pip, piperacillin; PipTzb, piperacillin–tazobactam

combination with one of the β-lactamase inhibitors, clavulanic acid or tazobactam, this mutant showed a smaller increase in resistance compared with exposure to mecillinam alone (2.6- and 4.8-fold, respectively), despite the concentration of mecillinam being the same in all three experiments (0.25 μg/ml at 1 × MIC for CTX-M-15WT). The single mutants CTX-M-15S133G and CTX-M-15G239S, which were initially selected on piperacillin–tazobactam, showed an increase in the MIC of

piperacillin–tazobactam by 8- and 4.5-fold (corresponding to 13 μg/ml and 7.3 μg/ml), compared with CTX-M-15WT (1.6 μg/ml) (Fig. 2a), thereby exceeding or approaching the clinical breakpoint for piperacillin–tazobactam of 8 μg/ml (as defined by EUCAST) (Supplementary Table 1).

Interestingly, for all double mutants (except CTX-M-15S133G-N135D) the MIC decreased to the levels of CTX-M-15WT falling below the clinical breakpoints for all the tested drugs. This result

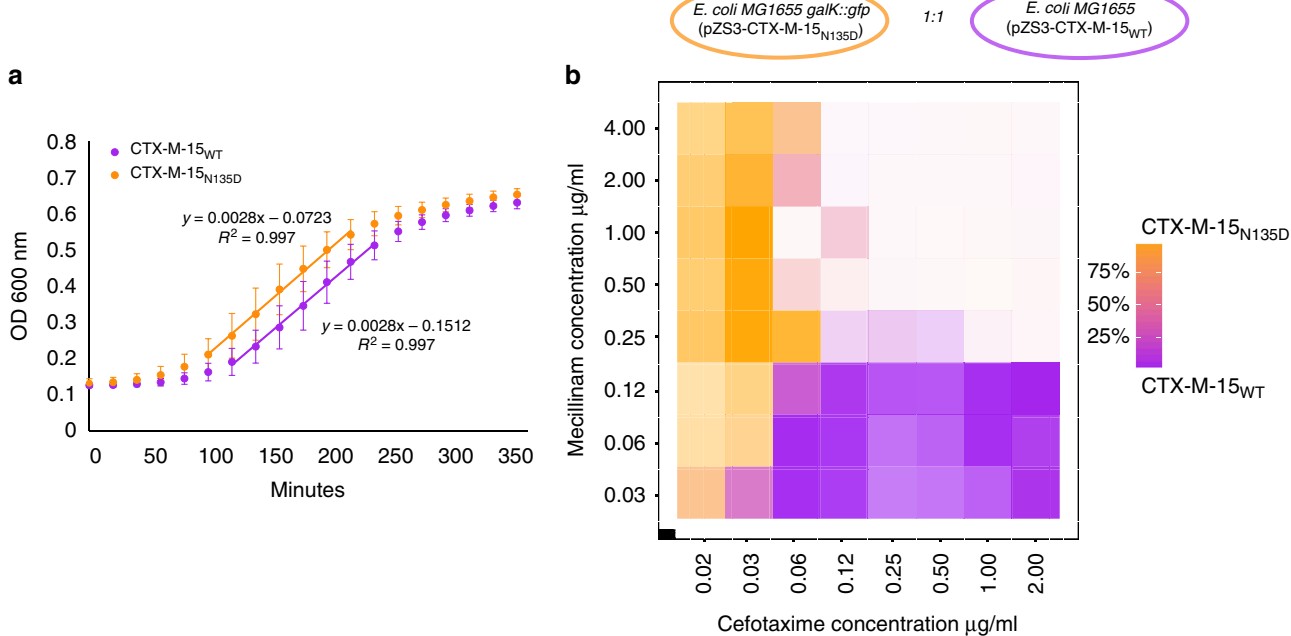

**Fig. 3** CTX-M-15WT vs. CTX-M-15N135D growth curves and 2D gradient plot. **a** *E. coli* TOP10 carrying either (pZS3-CTX-M-15WT) or (pZS3-CTX-M-15N135D) were grown in LB media in six biological replicates and growth rates were compared using linear regression on the exponential growth phase. No effect on the growth rate was observed for strains expression either enzyme in absence of antibiotic exposure. Error bars: SD. **b** *E. coli* MG1655 (pZS3-CTX-M-15WT) was mixed with *E. coli* MG1655 galK::gfp (pZS3-CTX-M-15N135D) in a 1:1 ratio and grown in a 2D gradient of mecillinam vs. cefotaxime. Each square represents a well and the colour represents the amount of either labelled CTX-M-15N135D (orange) or non-labelled CTX-M-15WT (purple). Darker colour indicates higher total cell count and lighter colour indicate lower cell count. The highest concentrations corresponded to mecillinam 4 × MIC for CTX-M-15WT and cefotaxime 2 × MIC for CTX-M-15N135D. At drug exposures above 0.06 μg/mL and 0.25 μg/mL of cefotaxime and mecilliam, respectively, neither strains expressing the CTX-M-15N135D or the CTX-M-15WT can survive

suggests the presence of negative epistasis between the mutations G239S and S133G or N135D. The CTX-M-15$_{S133G-N135D}$ double mutant had reduced resistance towards mecillinam compared with the mecillinam-selected CTX-M-15$_{N135D}$ single mutant but higher than that of the piperacillin–tazobactam-selected CTX-M-15$_{S133G}$ and CTX-M-15$_{WT}$. In addition, CTX-M-15$_{S133G-N135D}$ showed increased resistance against piperacillin–tazobactam compared with CTX-M-15$_{N135D}$, to a level comparable to that of the CTX-M-15$_{S133G}$ single mutant (Fig. 2b).

Collateral sensitivity to the cephalosporin cefotaxime was observed for all constructed single and double mutants. The cefotaxime MIC decreased from 23 μg/ml to < 2 μg/ml in all cases, which is below the clinical breakpoint of 2 μg/ml (as defined by EUCAST) (Supplementary Table 1).

Based on this data we wanted to examine whether the CTX-M-15$_{N135D}$ mutant had completely lost the ability to hydrolyse cephalosporins. As controls to test whether resistance levels in *E. coli* TOP10 cells were comparable to that of WT strains expressing CTX-M-15, we used *E. coli* MG1655 (K12) transformed with the plasmid (pZS3-CTX-M-15$_{WT}$) and a clinical *E. coli* ESBL strain, Ec35 (ST131), which carries the $bla_{CTX-M-15}$ WT resistance gene. Both strains showed high susceptibility to mecillinam and high resistance to all tested cephalosporins: cefuroxime (second generation), cefotaxime (third generation), and cefepime (fourth generation). In contrast, CTX-M-15$_{N135D}$-expressing *E. coli* MG1655 hydrolysed the three cephalosporins at low levels as reflected by the MICs, yet it showed high resistance towards mecillinam as expected (Fig. 2c). To ensure that the differences in susceptibilities between CTX-M-15$_{WT}$ and the CTX-M-15$_{N135D}$ mutant were not caused by differences in fitness, we measured the exponential growth rates for *E. coli* MG1655

(pCTX-M-15$_{N135D}$) and *E. coli* MG1655 (pCTX-M-15$_{WT}$). These strains had similar growth rates (Fig. 3a). These results demonstrate the occurrence of collateral sensitivity interactions within the CTX-M-15 gene between either piperacillin–tazobactam or mecillinam and cephalosporin class drugs.

**Mecillinam with cefotaxime selects against resistant mutants.** Based on the observed collateral sensitivity patterns, we hypothesized that a combination of a cephalosporin and either mecillinam or piperacillin–tazobactam could prevent the one-step evolution event leading to mecillinam or piperacillin–tazobactam resistance. To examine this idea, we investigated the combination of mecillinam and the cephalosporin cefotaxime. Mecillinam was chosen over piperacillin–tazobactam for several reasons; it is an important drug for treating urinary tract infections (mostly in Scandinavia); mecillinam works without addition of an inhibitor; it can reach very high concentrations in the urine (> 200 mg/l), which is desirable, as it is mainly used to treat urinary tract infections; it is well tolerated; has a high absorption rate in the intestine; a low occurrence of clinical resistance; and a minimal effect on gut and vaginal microflora[43].

To test whether any single mutants could confer resistance to the combination of mecillinam and cefotaxime, the $bla_{CTX-M-15}$ library was plated on agar plates containing both drugs. The concentrations were chosen such that mecillinam would select against CTX-M-15$_{WT}$ (4–16 × MIC for CTX-M-15$_{WT}$) and cefotaxime would select against mutants with decreased ability to hydrolyse cefotaxime (1–4 × MIC for CTX-M-15$_{N135D}$). Agar plates containing only mecillinam (8 × MIC for CTX-M-15$_{WT}$)

were used as a positive control. To ensure the representation of all possible CTX-M-15 variants, we plated approximately $6 \times 10^7$ cells providing $> 1000 \times$ coverage of possible mutants (see Methods). A few colonies grew on the combination plates but none were recovered after re-streaking, indicating that these colonies did not harbour genetically stable resistance to the drug combination. Ten colonies from the mecillinam-only selection plates were Sanger sequenced and all were verified to be CTX-M-15$_{N135D}$ single mutants. These results suggest that although mecillinam resistance can be effectively selected from the library, no single-nucleotide polyorphisms (SNPs) in the library provided combined mecillinam and cephalosporin resistance.

As CTX-M-15$_{N135D}$ was the mutant in the library showing the highest increase in mecillinam resistance, we wanted to explore how this mutant would grow in various concentrations of both mecillinam and cefotaxime. For this purpose, a two-dimensional (2D) competition experiment was performed with CTX-M-15$_{WT}$ and CTX-M-15$_{N135D}$ mixed in 1:1 ratio. This was performed to expose the the mecillinam-resistant mutant to a concentration gradient of the two drugs, having increased mecillinam concentrations on one axis and increased cefotaxime concentrations on the other axis. To be able to count the number of mutant cells using fluorescence-activated cell sorting (FACS), a strain of E. coli MG1655 was labelled with gfp (galK::gfp) and pZS3-CTX-M-15$_{N135D}$ was introduced. The two E. coli MG1655 strains were mixed at a 1:1 ratio and were grown for 5 h in a 2D gradient of mecillinam and cefotaxime (Fig. 3b). The frequency of both strains was assessed using flow cytometry and compared with the total number of cells in the respective well. We found that although the CTX-M-15$_{WT}$ carrying strain was highly enriched at high cefotaxime concentrations and the CTX-M-15$_{N135D}$ mutant at high mecillinam concentrations, the number of cells able to grow as both drug concentrations were increased was substantially reduced (Fig. 3b). These findings were confirmed in a repeated experiment with switched background genotypes (labelling of E. coli MG1655 (galK::gfp) (pZS3-CTX-M-15$_{WT}$) (Supplementary Fig. 1). This and the previous experiment show how collateral sensitivity between mecillinam and a cephalosporin limits the growth of mecillinam-resistant CTX-M-15 mutants.

**Mecillinam and cefotaxime inhibits growth of CTX-M-15 in mice.** Our in vitro results showed that a drug combination of mecillinam and cefotaxime could eliminate both cephalosporin-

resistant bacteria harbouring CTX-M-15$_{WT}$ and mecillinam-resistant bacteria harbouring the mutant CTX-M-15$_{N135D}$, thereby limiting the evolutionary potential of CTX-M-15. To test whether this would also be applicable in vivo, we used a well-established peritonitis mouse model system used to evaluate in vivo survival of the bacterium Salmonella enterica Typhimurium strain LT2 by assessing CFU counts in the liver and spleen. Salmonella was chosen due to its ability to spread to internal tissues making in possible to evaluate the level of infection based on CFU counts in the organs[44]. Mice were injected with bacteria carrying either CTX-M-15$_{WT}$ or the mecillinam-resistant CTX-M-15$_{N135D}$ variant on a plasmid. The infected mice were treated with mecillinam, cefotaxime, or a combination of mecillinam and cefotaxime (mecillinam, 100 mg/kg/day and cefotaxime, 150 mg/kg/day).

Mice infected with the strain expressing CTX-M-15$_{WT}$ and treated with mecillinam or a combination of mecillinam and cefotaxime had significantly lower bacterial counts in the liver than control mice and mice treated with cefotaxime. Conversely, mice infected with the strain expressing CTX-M-15$_{N135D}$ and treated with cefotaxime or a combination of cefotaxime and mecillinam had significantly lower counts in the liver than control mice and mice treated with mecillinam (Fig. 4a and b). CFU per gram counts were also evaluated in the spleen and showed similar results (Supplementary Fig. 2).

Importantly, the combination of mecillinam and cefotaxime significantly reduced the load of bacteria expressing either CTX-M-15$_{WT}$ or CTX-M-15$_{N135D}$ in both investigated organs. These data highlight the fact that the drug combination is effective in an animal infection model, strongly suggesting that the administration of mecillinam and cefotaxime together could potentially limit the evolution of the $bla_{CTX-M-15}$ gene, limiting both cephalosporin- and mecillinam-resistant variants.

## Discussion

In this study, we describe how collateral sensitivity patterns can arise in a horizontally transferred β-lactamase gene in response to resistance evolution. In particular, we find that CTX-M-15 variants that have increased resistance towards antibiotics, including mecillinam, become collaterally sensitive to cephalosporins. Based on these data we demonstrate in vitro that combination treatment with mecillinam and cefotaxime limits the ability of CTX-M-15 to counteract the antibiotics due to the reciprocal collateral sensitivity. Furthermore, we show that this drug combination can

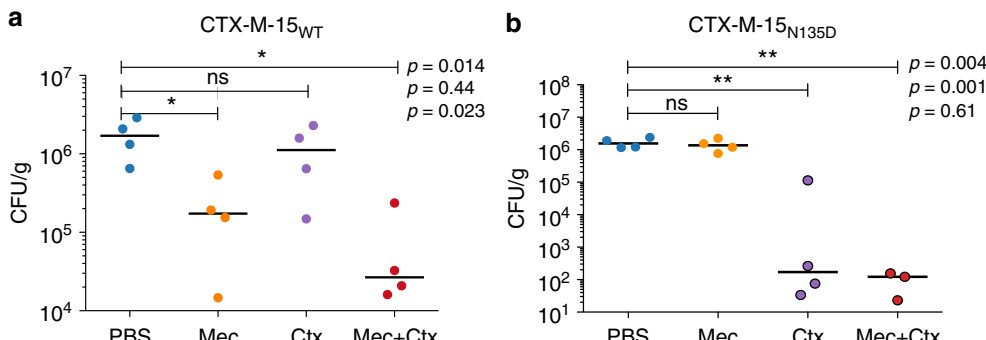

**Fig. 4** Bacterial counts from mice infected with CTX-M-15WT or CTX-M-15N135D after treatment. Mice infected via intraperitoneal injection with S. typhimurium expressing either CTX-M-15WT strain (**a**) or the mutant strain carrying CTX-M-15N135D (**b**) were treated with mecillinam (Mec), cefotaxime (Ctx), or a combination of the two. PBS was used as a negative control. Bacterial counts of liver tissue (colony-forming units (CFU)/g) are shown (four mice per group). Inoculum size was $1.5 \times 10^5$ CFU/mouse. ns: not significant; *$p < 0.05$; **$p < 0.01$. Horizontal bold line denotes the median. Statistical analysis was performed using unpaired Student's $t$-test. Symbols surrounded by a black border represent counts on plates with $< 20$ CFU (in those cases, statistical significance was calculated assuming 20 CFU/plate). One mouse infected with the mutant CTX-M-15N135D and treated with mecillinam in combination with cefotaxime cleared the infection. Data from this mouse are not plotted

overcome infections with strains expressing either of the two resistant variants of CTX-M-15, suggesting that the antibiotic combination could have a clinical utility.

Traditionally, collateral sensitivity interactions have been identified for chromosomal mutations using adaptive evolution[9,20,22,23,45,46]; yet, for horizontally acquired resistance genes collateral sensitivity has not been studied in-depth. One study found that amoxicillin–clavulanic acid adapted clones of *E. coli* expressing β-lactamases from the CTX-M family showed antagonistic pleiotropic effects between the β-lactam-inhibitor combinations piperacillin–tazobactam and amoxicillin–clavulanic acid, and several drugs within the cephalosporin class[33]. However, studying the evolutionary potential of a single acquired resistance gene using adaptive evolution can be challenging. For some antibiotics, including mecillinam, several chromosomal mutations in *E. coli* can lead to increased resistance. However, these mutations often incur increased fitness cost making them less clinically relevant[47]. Therefore, using traditional adaptive evolution to study mecillinam resistance conferred by horizontally transferred genes in *E. coli* is not feasible. To address this issue we examined the evolutionary potential and constraints of CTX-M-15 adaptive change using directed evolution, which enables the study of a single gene of interest without confounding factors resulting from host genome mutations in response to antibiotic exposure[31,32]. This laboratory approach is a simplification compared with the clinical setting in which both chromosomal mutations and acquired resistance genes contribute to resistance phenotypes in response to antibiotic gradients in space and time. However, we consider the approach relevant, as it allows an assessment of the evolutionary potential of acquired resistance genes under different selective conditions.

The CTX-M-15 enzyme proved to be very sensitive to mutations in the active site, leading to loss of cephalosporin resistance, which might explain the low level of mecillinam and piperacillin–tazobactam resistance generally observed within the CTX-M-1 and CTX-M-9 families[33,48]. Interestingly, this type of mutually exclusive resistance patterns has also been observed for other resistance genes, e.g., aminoglycoside-modifying enzymes[49]. This indicates that it is difficult for antibiotic resistance enzymes to evolve an active site that efficiently hydrolyses several distinct types of drugs. The prevalence of such collateral sensitivity interactions strongly warrants further research into how acquired antibiotic resistance genes can be countered using drug combination therapy. Even exploiting the differential sensitivity of related drugs from the same class could be a promising avenue for drug combination therapy. Although the screening approach deployed in this study could be used to identify new drug combinations, structural studies at molecular resolution of acquired resistance genes could also help to elucidate their evolutionary potential and adaptive constraints.

In conclusion, antibiotic combinations, even of related drugs, show potential for counteracting the rapidly developing problem of acquired antibiotic resistance. In-depth studies of new drug combinations should be initiated both in the laboratory as well as in the clinic, specifically for dealing with the global problems of ESBLs. In particular, the drug combination of mecillinam and cefotaxime warrants further attention for management of urinary tract infections (UTIs) caused by ESBLs. Both due to the high efficacy of mecillinam in treating UTIs and because the levels of mecillinam resistance in the clinic is still low[42]. Future in vivo and human studies should illuminate whether this antibiotic combination approach based on collateral sensitivity could help address the issue of antibiotic resistance in the clinic. The approach employed in this study may extend the use of existing or novel antibiotics and help to predict the evolutionary directions of emerging resistance genes.

## Methods

**Construction of the *bla*CTX-M-15 mutant library**. As a backbone for the *bla*CTX-M-15 mutant library we used the low-copy number kanamycin-resistant plasmid pZS24 (region 678–3984) from the pZ-vector system (http://www.expressys.com/main_vectors.html). A geneblock (IDT) containing the β-lactamase promoter, terminator, and multiple cloning sites was blunt-end cloned into the backbone and the resulting plasmid was named pZS3. Solution was treated with DpnI (ThermoFisher #FD1703) and purified using QIAquick PCR purification kit (Qiagen). The *bla*CTX-M-15 gene was amplified from a plasmid isolated from an *E. coli* clinical isolate using the primers CTX-M-15-F and CTX-M-15-R (primer sequence in Supplementary Table 4). The primers contained restriction enzyme tails for SalI and HindIII also present in the geneblock. pZS3 and purified CTX-M-15 PCR product (PCR purification kit, Qiagen) were mixed with each restriction enzyme (Thermo Fisher #FD0644 and #FD0505) + Fast Digest Buffer 10 × (ThermoFisher #B64) according to manufacturer's protocol. Solution was purified with QIAquick PCR purification kit (Qiagen). The two products were ligated using T4 DNA ligase and ligase buffer (ThermoFisher #EL0011). The resulting plasmid was named pSZ3-CTX-M-15WT. To create a backbone that could be used for error-prone PCR and where negative selection against the *bla*CTX-M-15 WT gene could be performed, we used the plasmid pJET containing an ampicillin resistance gene (ThermoFisher #K1231). *bla*CTX-M-15 was amplified from pZS3-CTX-M-15WT using the primers pZ-insert-F and CM-insert-R pZS. DNA ligase and T4 DNA ligase buffer (ThermoFisher #EL0011) were used during the cloning procedures. This plasmid was named pJET-CTX-M-15WT. Then, error-prone PCR was performed with the Genemorph II random mutagenesis kit (Agilent Technologies #200550) run on the *bla*CTX-M-15 WT gene with pJET-CTX-M-15WT as a template (500 ng of target DNA) following the manufacturer's protocol using the primers gblok-CTX-M-15-F and gblok-CTX-M-15-R. The PCR product run on an agarose gel and the band was cut and purified with a gel-extraction kit (Qiagen). The error-prone PCR product was then used as a template for PCR with primers that added restriction enzyme tails (AscI-CTX-M-15-F and PmeI-CTX-M-15-R) during 20 PCR cycles. The PCR product was run on an agarose gel and purified with the Gel-Extraction Kit (Qiagen). To amplify a backbone for the mutant CTX-M-15 library, we used pZS3 as a template and primers containing restriction enzyme tails for AscI and PmeI: PmeI-Ancestral-pZS3-F and AscI-Ancestral-pZS3-R. Both products were cut (ThermoFisher enzymes: #ER1891 and #ER1341) and purified using QIAquick PCR purification kit (Qiagen), and dephosporylated using 1 μl Phosphatase fast digest (ThermoFisher #EF0654) and purified with QIAquick PCR purification (Qiagen). Ligation was performed using vector:insert ratio of 1:10 and ligated using T4 DNA ligase and ligase buffer (ThermoFisher #EL0011). The product resulting from this ligation, containing the *bla*CTX-M-15 SNP library named pZS3-CTX-M-15-Mut2Vol2, was transformed into electrocompetent *E. coli* TOP10 (Thermo-Fisher #C66455) and selected on kanamycin (50 μg/ml). The second library was constructed similarly to the first library, except that 8 instead of 20 cycles were run on the error-prone PCR product with primers containing restriction enzyme tails. The second library was named pZS3-CTX-M-15-Mut6. The complete sequence of the *bla*CTX-M-15 gene and the β-lactamase promoter can be found in Supplementary Data 1.

**Estimation of the library coverage**. The *bla*CTX-M-15 gene contains 875 bp, which leaves $875 \times 3 = 2625$ total possible SNPs of the gene. In total, 1153 *bla*CTX-M-15 SNPs (43% of the 2625 possible SNPs) were detected after sequencing, with frequencies above the Illumina sequencing error rate, which is estimated to be ~0.1%[50]. In total, the two libraries comprised $2 \times 10^6$ clones based on an average CFU count from dilution count assay. Based on Sanger sequencing of 24 colonies, the average mutation rate was estimated to be 0.957 (8 with 0 mutations, 10 with 1 mutation, 4 with 2 mutations and 1 with 4 mutations). Using the PEDEL programme that estimates the diversity in error-prone PCR libraries, http://guinevere.otago.ac.nz/cgi-bin/aef/pedel.stats.pl[51], the estimated number of mutants with exactly one mutation in both libraries was found to be $7.351 \times 10^5$. This renders a SNP coverage of $735{,}100/2625 = 280$, assuming equal distribution of all SNPs.

**Screening of the mutant library on a variety of antibiotics**. The CTX-M-15-Mut2Vol2 library was screened on LB agar plates containing the antibiotics shown in the Table 1. CTX-M-15WT was screened on the same plates as control. The second library Mut6 was screened in the same way and on additional antibiotics: ertapenem, amoxicillin–clavulanic acid, and mecillinam in combination with the two inhibitors tazobactam and clavulanic acid (SM).

To ensure that all SNPs were present on each plate at least once, the coupon collector problem[52] was applied, as follows:

$$M = N \frac{\log(-N)}{\log(P)} \quad (1)$$

For $P = 99.9\%$:

$$M = 2625 \frac{\log(-2625)}{\log(0.999)} = 39.000 \text{ cells}$$

**Table 1 Concentrations of antibiotics used for screening the CTX-M-15 library**

| MIC | PIP-TZB | Mecillinam | Meropenem | Ceftazidime |
|-----|---------|------------|-----------|-------------|
| 0.5 | 4 | 0.125 | 0.03 | — |
| 0.75 | 8 | 0.1875 | 0.045 | — |
| 1 | 16 | 0.25 | 0.06 | — |
| 2 | 32 | 0.5 | 0.12 | 4 |
| 4 | 64 | 1 | 0.24 | 8 |

Concentrations are in μg/ml. PIP-TZB: piperacillin–tazobactam. Ceftazidime: positive control

where $M$ is the number of cells on each plate, $N$ is total SNPs and $P$ is the probability of obtaining all SNPs. The number of cells should be as low as possible to avoid overgrowth but high enough to ensure the presence of all mutants with a probability of 99.9%. Assuming an error rate of $10/24 \times 100 = 42\%$ of all cells to contain SNPs (based on Sanger sequencing), at least $39,000$ cells $\times (100/42) = 92,857$ cells should be plated on each plate to obtain 99.9% of all SNPs in the library. In total, we plated $\sim6 \times 10^7$ cells for a very high probability of plating all clones. One hundred microlitres were plated on each plate, the mutant library in four replicates and CTX-M-15$_{WT}$ in two replicates.

**Next-generation sequencing**. Next-generation sequencing (NGS) was performed using the MiSeq System from Illumina for the $0.5–4 \times$ MIC libraries selected on either piperacillin–tazobactam or mecillinam, the single library selected on $0.5 \times$ MIC meropenem and one library grown on LB agar without antibiotics used as control (for the library CTX-M-15-Mut2Vol2). The library CTX-M-15-Mut6 was screened in the same way. DNA was isolated by adding 2 ml of 0.9% NaCl to each plate and scraping off the colonies. Two hundred microlitres of the solution were centrifuged for 4 min. Plasmid purification was performed with the QIAprep Spin Miniprep Kit (Qiagen) according to the manufacturer's recommendations. The Nextera XT DNA kit (Illumina) was used for library preparation. CLC Genomic Workbench was used to transform the Illumina sequencing data. A cutoff SNP frequency value of 0.1% was used, owing to the Illumina sequencing error rate[50].

**Construction of single and double CTX-M-15 mutants**. Primers introducing the mutations A > G at position 397 (S133G), A > G at position 403 (N135D) and G > A at position 715 (G239S) were used to amplify the $bla_{CTX-M-15}$ WT gene using pJET-CTX-M-15$_{WT}$ as a template (primer sequences in Supplementary Table 4). To amplify a backbone for the mutants, we used pZS3 as a template, and primers Ancestral-pZS3-F and Ancestral-pZS3-R. PCR product was DnpI treated (ThermoFisher #FD1703) and purified using QIAquick PCR purification kit (Qiagen). The amplicon was blunt-end cloned using T4 DNA ligase and ligase buffer (ThermoFisher #EL0011). Ligation solution was purified with PCR cleanup (A&A Technologies). Each of the resulting plasmids pZS3-CTX-M-15$_{N135D}$, pZS3-CTX-M-15$_{S133G}$ and pZS3-CTX-M-15$_{G239S}$ were transformed into electrocompetent E. coli TOP10 (ThermoFisher #C66455). Mutations were verified with Sanger sequencing. The double mutants S133G-G239S and S133G-N135D were constructed using $bla_{CTX-M-15\ S133G}$ as template DNA for PCR with primers introducing the mutations G239S and N135D, respectively. The double mutant N135D-G239S was constructed using $bla_{CTX-M-15\ N135D}$ as template. Same cloning procedure was performed as for the single mutants.

**MIC tests for single and double CTX-M-15 mutants**. MIC tests were performed for each of the six pZS3-CTX-M-15 mutants, the pZS3-CTX-M-15$_{WT}$ and pZS3 as a control. MHBII media was used for MIC tests (Sigma). Each strain was inoculated separately in kanamycin-MHBII media (10 μg/ml) and grown overnight to a cell density of $10^{10}$ CFU/ml. The cells were diluted in MHBII media to a concentration of $10^8$ CFU/ml. Ninety-six deep-well plates were used to prepare the antibiotic concentrations and 150 μl were copied into five 96-well plates. One microlitre of the $10^8$ dilutions was inoculated into a twofold antibiotic gradient in liquid media in five technical replicates and grown with shaking at 37 °C overnight. Growth was measured using spectrophotometry. The MIC values were determined as the concentration at which cell growth was 100% inhibited (OD corresponds to pZS3). The following antibiotics and concentrations (lowest to highest of twofold dilution assay) were used: mecillinam (Sigma, CASRN: 32887-01-7, assigned by the Chemical Abstracts Service) (0.0156–4 μg/ml), amoxicillin trihydrate (TCI Chemicals, CASRN:61336-70-7), potassium clavulanate (Fluka Analytic, CASRN:61177-45-5), cefotaxime sodium salt (Sigma, CASRN:64485-93-4), meropenem (Astra-Zeneca, CASRN:119478-56-7) (0.00375–0.96 μg/ml), piperacillin sodium salt (Sigma, CASRN:59703-84-3) (16–4096 μg/ml) and tazobactam (TCI Chemicals, CASNR:89786-04-9). Combinations of inhibitor and antibiotic were mixed according to the clinical usage of piperacillin–tazobactam (8:1) (total concentration of twofold dilutions: 0.25–64 μg/ml) and amoxicillin–clavulanic acid (4:1) (total concentration of twofold dilution: 0.5–128 μg/ml). The concentrations

of tazobactam and clavulanic acid with mecillinam are consistent with the inhibitor concentrations in piperacillin–tazobactam and amoxicillin–clavulanic acid. Mecillinam–tazobactam = 1:2 (0.0156/0.0275–4/7.04 μg/ml) and mecillinam–clavulanic acid = 1:6.4 (0.0156/0.1–4/25.6 μg/ml). E. coli MG1655 was transformed with the plasmid pZS3-CTX-M-15$_{WT}$ and MIC tests were performed against mecillinam and three different cephalosporins: cefuroxime (Sigma, CASRN:56238-63-2), cefotaxime and cefepime (CASRN:88040-23-7) (fourth generation). The clinical E. coli ESBL strain ST131 carrying CTX-M-15 was used to validate similar resistance levels between this strain, E. coli MG1655 and E. coli TOP10.

**Growth experiment in a gradient of mecillinam vs. cefotaxime**. E. coli MG1655 (pZS3-CTX-M-15$_{WT}$) was mixed with E. coli MG1655 galK::gfp (pZS3-CTX-M-15$_{N135D}$) (constructed by inserting gfp within the galK gene) in a 1:1 ratio and grown in a 2D gradient of mecillinam vs. cefotaxime. Cells were grown for 5 h at 37 °C with shaking in LB medium and 10 μl of the cell suspension was inoculated into each well (five technical replicates were used). The cefotaxime concentrations were 0.02–2 μg/ml and mecillinam concentration was 0.03–4 μg/ml. The well with the highest concentration of antibiotics corresponded to CTX-M-15$_{WT}$ 4 × MIC for mecillinam and CTX-M-15$_{N135D}$ 2 × MIC for cefotaxime. The experiment was performed again with E. coli MG1655 galK::gfp (pZS3-CTX-M-15$_{WT}$) against E. coli MG1655 (pZS3-CTX-M-15$_{N135D}$), to ensure that green fluorescent protein expression did not affect the experimental outcome. Cells were counted by FACS sorting.

**Growth rate experiment**. E. coli MG1655 (pZS3-CTX-M-15$_{WT}$) and E. coli MG1655 (pZS3-CTX-M-15$_{N135D}$) were grown overnight in 96-well plates in 200 μl LB with kanamycin (50 μg/ml), using six biological replicates. A new 96-well plate with 200 μl LB without antibiotic was used to inoculate 1 μl of the overnight cultures. The negative control wells contained no cells. The plate was covered with Breath-Easy seal and a kinetic run was performed in a plate reader (BioTek, Elx808). Linear regression was performed on the data points from the exponential phase with Microsoft Excel.

**Mouse study**. Fresh overnight cultures of S. typhimurium strain LT2 carrying pZS3-CTX-M-15$_{WT}$ or pZS3-CTX-M-15$_{N135D}$ were grown in LB medium at 37 °C with shaking at 190 r.p.m. until saturation. The cells were washed three times in sterile phosphate-buffered saline (PBS). Female BALB/c mice (6 weeks old) were used (Charles River Laboratories). The mice were injected intraperitoneally with 100 μl of ~$10^6$ CFU/ml bacteria. The inoculum size was quantified by plating appropriate tenfold dilutions on LB agar plates supplemented with kanamycin (50 μg/ml) to select for the pZS3 plasmid. Thirty minutes post infection, the mice were divided into four treatment groups (four mice/group) and treated intraperitoneally (50 μl) with PBS, mecillinam (100 mg/kg/day), cefotaxime (150 mg/kg/day) or a combination of both drugs. Antibiotics were administered in two half-daily doses for 36 h. Two days after the first treatment dose, the mice were killed by cervical dislocation. The spleens and livers were collected and homogenized cell suspensions were prepared in PBS. Tenfold dilutions were spread on LB plates supplemented with kanamycin, to determine the bacterial burden in mice. The bacterial counts were normalized per gram of tissue (CFU/g). Uppsala Animal Ethics Committee (permit no C154/14) approved all mouse experiments, which were performed in accordance with national and institutional guidelines in the Swedish National Veterinary Institute in Uppsala, Sweden.

**Statistical analysis**. The inhibitory concentration for CTX-M-15WT and constructed mutant strains were determined as the lowest drug concentration that inhibited 100% growth compared with control based on $A_{600}$ measurements (IC$_{100}$), using five replicates. Unpaired Student's $t$-test was used for statistical analysis of difference between mice groups based on CFU/g from the liver and spleen, using four replicates (*$p < 0.05$, **$p < 0.01$). The coupons collector problem was applied to calculate how many mutant library clones should be plated on a selective plate to have all clones represented with 99.9% probability, using equation 1.

**Construction of a second CTX-M-15 library**. The second library was made using the error-prone PCR library also used to construct CTX-M-15-Mut2Vol2. The method was identical to the construction of the first library, except that 8 instead of 20 cycles were run with restriction enzyme tails when amplifying the error-prone PCR. The second mutant library was plated on a non-selective LB plate and the following antibiotics and antibiotic-inhibitor combinations piperacillin and tazobactam, mecillinam, mecillinam–tazobactam, mecillinam–clavulanic acid, meropenem, ertapenem (Merck, CASRN: 1-9-1), amoxicillinclavulanic acid and ceftazidime (TCI Chemicals, 72558-82-8) (negative control). NGS was then performed on the growth results. The CTX-M-15-Mut6 library contained 895 unique SNPs with a frequency of > 0.1%. In total, 1153 non-redundant SNPs were found in the two combined libraries.

**Reporting summary**. Further information on experimental design is available in the Nature Research Reporting Summary linked to this article.

## Data availability

All relevant raw data are available from the authors upon request.

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

## Acknowledgements

We acknowledge funding from The European Union H2020 (ERC-2014-STG) under Grant Agreement 638902, LimitMDR, Danish Council for Independent Research, Sapere Aude Programme DFF 4004-00213, the Lundbeck Foundation under grant agreement R140-2013-13496 and The Novo Nordisk Foundation under NFF grant number NNF10CC1016517, and The Swedish Research Council (DIA) grant number 2017-01527.

## Author contributions

M.O.A.S. and C.M conceived the original idea. M.O.A.S and C.M. and C.R designed the experiments. C.R. and C.M performed the experiments and analysed the data excluding the in vivo study. D.A., M.O.A.S., M.L. and C.M. designed the in vivo study. A.P. and M.L. performed the in vivo study and A.P., M.L. and D.A. analysed the in vivo study data. C.R. wrote the manuscript. All other authors commented on the manuscript.

## Additional information

**Competing interests:** The authors declare no competing interests.

