## [Peer Review File · Nature Communications]

REVIEWERS' COMMENTS:

Reviewer #1 (Remarks to the Author):

Review of the manuscript NCOMMS-18-32671

The manuscript entitled "Collateral sensitivity constrains the resistance evolution of the CTX-M-15 β -lactamase", submitted to Nature Communications by Carola E. H. Rosenkilde, as first author and Morten Sommer as corresponding author (NCOMMS-18-32671), explores the antibacterial potential of antibiotic combinations, which show collateral susceptibility phenomena (negative epistasis among the mutations involved in the resistance to one or another antibiotic), using in this case cefotaxime (CTX) and mecillinam (MEC) against infections caused by microorganisms carrying CTX-M-15 beta-lactamase as mechanism of resistance. They combine the 2D gradient, NGS technology and murine model to predict the difficulties to find resistant-variants. In fact, they confirm the clearance of *Salmonella enterica* Typhimurium strain LT2 carrying CTX-M-15 (cefotaxime-resistant pattern) and CTX-M-15N135D (mecillinam-resistant pattern) in artificially infected mice. This manuscript is an interesting work, because the authors propose us to take a step forward to combat the antibiotic resistance problem, based on the previously described collateral susceptibility phenomena. I would like to suggest the authors several unresolved questions.

The strategy of combined therapies based on collateral susceptibility phenomena is an attractive proposal to reduce the adaptive possibilities of enzymatic mechanisms of resistance (in this case a beta-lactamase). However, mutations in other parts of the genome could contribute to the resistance phenotype without changes in the beta-lactamase. In laboratory conditions the MEC-resistance phenotype is easily observed due to high number of potential targets. In fact, the most frequently MEC-r mutant recovered shows a chromosomal mutation in the *cysB* gene encoding for the CysB protein. Therefore, the success of the strategy based on collateral susceptibility would have more probabilities using combinations of antibiotics with low number of "clue point of resistance". Therefore, the authors could describe us if they found bacterial clones growing in presence of CTX and MEC simultaneously, although these mutants showed a CTX-M-15 without changes; in other words, what proportion in each well of 2D gradient was found? I do not know if the authors performed the same in vitro experiment using CTX and PTZ simultaneously (2D gradient), but it could help us to learn the best combinations.

Did not the authors found, -thanks to NGS technology-, mutants conferring low-level resistance to MEC? In these cases, the collateral susceptibility is lower striking, and consequently the mutations of second order are easier to be selected conferring resistance during prolonged treatment. Moreover, I would like to ask about the compensatory mutations restoring partially or completely the collateral susceptibility. In the case of *E. coli* carrying CTX-M beta-lactamases growing in presence of CTX and PTZ, other authors found that L169S mutation partially restored the effect of S130G mutation (or S133G in your numbering). Could be selected in vivo both low-level resistance mutants as compensatory mutations if the antibiotics used have not similar biodisponibility and pharmacokinetic profiles?

The murine model experiments suggest that MEC-resistant clones could be selected easily in monotherapy. Did you analyse these mutants? Moreover, as you know, only 25% of active metabolite of CTX is secreted in the urine. Then, is it enough to avoid the monotherapy in the combination treatment proposed by the authors for urine tract infections?

On the other hand, in vitro experiments showed that the selection of resistant variants was higher in presence of PTZ and MEC; it is surprising. Although, changes in CTX-M involved in MEC-r phenotype are lower than PTZ; however, the bacterial MEC-r variants are easier than PTZ.

In Table S1, could be a mistake the MIC value for MEC? In main text, line 153 the authors wrote 15 mcg/mL, whereas in table S1 wrote 8. Moreover, in table S2, the authors did not find clones at MIC 2-4-fold MIC; however, the CTX-M-15N135D mutant was also recovered at these concentrations. Probably, I misunderstood the results, but I would thank a clearest explanation.

Reviewer #2 (Remarks to the Author):

This manuscript is an outstanding study of evolutionary potential applied to a clinical problem. The authors first selected for highly resistant mutants of the CTX-M-15 resistance gene and identified novel, highly resistant mutants. Then they showed that collateral sensitivity happens between different antibiotics (Mecillinam and cephalosporins). Then they showed that co-application of those antibiotics reduced the evolution of resistance. Then they confirmed these results in mice.

I have read this study four times through trying to find technical, verbal or logical points to criticize. I have been thorough and I don't have any. If I could sit with the authors and talk to them about this work, I would point out that the mutations they identified might never be selected because they used single antibiotic selection in the beginning, but any concerns I had about that were gone by the end when I saw the combination therapy results where cefotaxime was used as a continued selective pressure.

The other point I would discuss with the authors is that there is work currently underway by Arjan de Visser to determine the stochasticity of selection outcomes from libraries. Different results may emerge in different rounds of selection. That doesn't change the conclusions of this research or negatively impact the findings, but in future experiments, it would be good to keep in mind.

In summary, I find this paper to be a complete delight to read. It is clear, thorough, expansive in scope, relevant, and exciting.

Reviewer #1 (Remarks to the Author):

Review of the manuscript NCOMMS-18-32671

The manuscript entitled “Collateral sensitivity constrains the resistance evolution of the CTX-M-15 β -lactamase”, submitted to Nature Communications by Carola E. H. Rosenkilde, as first author and Morten Sommer as corresponding author (NCOMMS-18-32671), explores the antibacterial potential of antibiotic combinations, which show collateral susceptibility phenomena (negative epistasis among the mutations involved in the resistance to one or another antibiotic), using in this case cefotaxime (CTX) and mecillinam (MEC) against infections caused by microorganisms carrying CTX-M-15 beta-lactamase as mechanism of resistance. They combine the 2D gradient, NGS technology and murine model to predict the difficulties to find resistant-variants. In fact, they confirm the clearance of *Salmonella enterica* Typhimurium strain LT2 carrying CTX-M-15 (cefotaxime-resistant pattern) and CTX-M-15N135D (mecillinam-resistant pattern) in artificially infected mice. This manuscript is an interesting work, because the authors propose us to take a step forward to combat the antibiotic resistance problem, based on the previously described collateral susceptibility phenomena. I would like to suggest the authors several unresolved questions.

The strategy of combined therapies based on collateral susceptibility phenomena is an attractive proposal to reduce the adaptive possibilities of enzymatic mechanisms of resistance (in this case a beta-lactamase). However, mutations in other parts of the genome could contribute to the resistance phenotype without changes in the beta-lactamase. In laboratory conditions the MEC-resistance phenotype is easily observed due to high number of potential targets. In fact, the most frequently MEC-r mutant recovered shows a chromosomal mutation in the *cysB* gene encoding for the CysB protein. Therefore, the success of the strategy based on collateral susceptibility would have more probabilities using combinations of antibiotics with low number of “clue point of resistance”. Therefore, the authors could describe us if they found bacterial clones growing in presence of CTX and MEC simultaneously, although these mutants showed a CTX-M-15 without changes; in other words, what proportion in each well of 2D gradient was found? I do not know if the authors performed the same in vitro experiment using CTX and PTZ simultaneously (2D gradient), but it could help us to learn the best combinations.

Answer: Thank you for your review and comments to our paper. Below we have addressed the specific comments and questions.

This is an interesting comment. However, we did not look for chromosomal changes in the cells that were detected at a high concentration of both mecillinam and cefotaxime. There was a low cell count but we believe that this was background from the FACS and that these cells did not in fact grow (see Figure 3 below that shows the cell count

The combination of mecillinam and cefotaxime would not prevent mecillinam resistance to occur based on chromosomal mutations, but since it is well established that chromosomal mecillinam resistance mutations have a higher fitness cost for the cell, they are not as readily selected for (e.g. the *cysB* mutation (Thulin, 2015, DOI: 10.1128/AAC.04819-14) This is in contrast to the CTX-M-15-N135D mutation which showed no difference in fitness cost compared to the WT CTX-M-15.

Did not the authors find, -thanks to NGS technology-, mutants conferring low-level resistance to MEC? In these cases, the collateral susceptibility is lower striking, and consequently the mutations of second order are easier to be selected conferring resistance during prolonged treatment. Moreover, I would like to ask about the compensatory mutations restoring partially or completely the collateral susceptibility. In the case of *E. coli* carrying CTX-M beta-lactamases growing in presence of CTX and PTZ, other authors found that L169S mutation partially restored the effect of S130G mutation (or S133G in your numbering). Could be selected in vivo both low-level resistance mutants as compensatory mutations if the antibiotics used have not similar biodisponibility and pharmacokinetic profiles?

Answer: These are interesting ideas and experiments to explore but they are outside the scope of this study. It would indeed be interesting to use the low level resistant mutants as a starting point for an adaptive evolution experiment. However, the problem with the high number of chromosomal mutations causing increased mecillinam resistance would most likely still interfere with the experiment.

We did find a recent study showing that the double mutation: S133T and A80V rendered CTX-M-15 double resistant to both cefotaxime and piperacillin-tazobactam (shen_2017, DOI: 10.1128/AAC.01848-16). However, none of these single mutations (based on data from our dataset and Shen et. al) individually gave increased resistance to piperacillin-tazobactam. The probability of two mutations occurring simultaneously is much lower than for single mutations (i.e. the product of the individual frequencies) and it is unlikely that they both appear simultaneously.

It is possible that for other drugs CTX-M-15 would be more flexible with regards to mutating in single positions without getting susceptible to cephalosporins. It would be interesting to perform more tests on this, since it seems that the CTX-M-15 active site is so specialised in binding cephalosporins that any change in the active site destroys this ability.

The murine model experiments suggest that MEC-resistant clones could be selected easily in monotherapy. Did you analyse these mutants? Moreover, as you know, only 25% of active metabolite of CTX is secreted in the urine. Then, is it enough to avoid the monotherapy in the combination treatment proposed by the authors for urine tract infections?

Answer: We plated the bacterial clones from the animals after treatment on mecillinam or cefotaxime to analyse the proportion of resistant clones (e.g. were there any CTX-M-15-N135D clones left after treatment with mecillinam and vice versa). Some clones were present on these plates but we did not analyse further their genetic background.

The second question, regarding the dose of cefotaxime in combination with mecillinam is certainly relevant and could be tested on a UTI model. However, the mutant: CTX-M-15-N135D is highly susceptible and MIC is $\ll 2\mu\text{g/ml}$ (see 2D plot above where proportion of N135D at $0.25\mu\text{g/ml}$ is close to zero), and since the clinical breakpoint for cefotaxime is $2\mu\text{g/ml}$, it should be possible to obtain a high enough concentration in urine even if only 25% of the metabolite is secreted in the urine.

On the other hand, in vitro experiments showed that the selection of resistant variants was higher in presence of PTZ and MEC; it is surprising. Although, changes in CTX-M involved in MEC-r phenotype are lower than PTZ; however, the bacterial MEC-r variants are easier than PTZ. In Table S1, could be a mistake the MIC value for MEC? In main text, line 153 the authors wrote 15 mcg/mL , whereas in table S1 wrote 8. Moreover, in table S2, the authors did not find clones at MIC 2-4-fold MIC; however, the CTX-M-15N135D mutant was also recovered at these concentrations. Probably, I misunderstood the results, but I would thank a clearest explanation.

Answer: Sorry about this. The concentration in Supplementary Table 1 should be 15 and not 8. It has been corrected.

With regards to Supplementary Table 2 it is correct that we did not find any clones at 2-4 x MIC. We believe that this is due to uncertainties of the antibiotic concentrations when mixing the selection plates such that the concentrations of mecillinam in the plates are in fact higher than the specified MIC. That was why we explicitly tested the MIC again for the mutants after selecting them on the plates.

Reviewer #2 (Remarks to the Author):

This manuscript is an outstanding study of evolutionary potential applied to a clinical problem. The authors first selected for highly resistant mutants of the CTX-M-15 resistance gene and identified novel, highly resistant mutants. then they showed that collateral sensitivity happens between

different antibiotics (Mecillinam and cephalosporins). Then they showed that co-application of those antibiotics reduced the evolution of resistance. Then they confirmed these results in mice

I have read this study four times through trying to find technical, verbal or logical points to criticize. I have been thorough and I don't have any. If I could sit with the authors and talk to them about this work, I would point out that the mutations they identified might never be selected because they used single antibiotic selection in the beginning, but any concerns I had about that were gone by the end when I saw the combination therapy results where cefotaxime was used as a continued selective pressure.

The other point I would discuss with the authors is that there is work currently underway by Arjan de Visser to determine the stochasticity of selection outcomes from libraries. Different results may emerge in different rounds of selection. That doesn't change the conclusions of this research or negatively impact the findings, but in future experiments, it would be good to keep in mind.

In summary, I find this paper to be a complete delight to read. It is clear, thorough, expansive in scope, relevant, and exciting.

Answer: Thank you for your comments and positive review of our paper. We have thought much about the size of our library, and how to estimate the correct library size to ensure that all possible single mutations are present. It is of course very relevant to consider the possibility that another mutational library could give different results. That was why we did do the screening on two different libraries, but overall, we saw the same results. Still, it is possible that a new library would yield other resistant clones not detected in the first two rounds and that is relevant to consider. However, as you mention we did find a mecillinam clone showing collateral sensitivity, which is a very interesting observation. But it is possible that there are other SNPs of CTX-M-15 with the same characteristics.